# Supporting Tremor Rehabilitation Using Optical See-Through Augmented Reality Technology

**DOI:** 10.3390/s23083924

**Published:** 2023-04-12

**Authors:** Kai Wang, Dong Tan, Zhe Li, Zhi Sun

**Affiliations:** 1School of Art and Design, Wuhan University of Technology, Wuhan 430070, China; wkaizh@gmail.com (K.W.);; 2Graduate School of Engineering Science, Osaka University, Toyonaka 5608531, Japan; 3College of Education, Fujian Normal University, Fuzhou 350117, China; 4Graduate School of Human Sciences, Osaka University, Suita 5650871, Japan

**Keywords:** optical see-through augmented reality, rehabilitation, tremor, yapa-PBGA, movement sensing

## Abstract

Tremor is a movement disorder that significantly impacts an individual’s physical stability and quality of life, and conventional medication or surgery often falls short in providing a cure. Rehabilitation training is, therefore, used as an auxiliary method to mitigate the exacerbation of individual tremors. Video-based rehabilitation training is a form of therapy that allows patients to exercise at home, reducing pressure on rehabilitation institutions’ resources. However, it has limitations in directly guiding and monitoring patients’ rehabilitation, leading to an ineffective training effect. This study proposes a low-cost rehabilitation training system that utilizes optical see-through augmented reality (AR) technology to enable tremor patients to conduct rehabilitation training at home. The system provides one-on-one demonstration, posture guidance, and training progress monitoring to achieve an optimal training effect. To assess the system’s effectiveness, we conducted experiments comparing the movement magnitudes of individuals with tremors in the proposed AR environment and video environment, while also comparing them with standard demonstrators. Participants wore a tremor simulation device during uncontrollable limb tremors, with tremor frequency and amplitude calibrated to typical tremor standards. The results showed that participants’ limb movement magnitudes in the AR environment were significantly higher than those in the video environment, approaching the movement magnitudes of the standard demonstrators. Hence, it can be inferred that individuals receiving tremor rehabilitation in the AR environment experience better movement quality than those in the video environment. Furthermore, participant experience surveys revealed that the AR environment not only provided a sense of comfort, relaxation, and enjoyment but also effectively guided them throughout the rehabilitation process.

## 1. Introduction

A tremor is a type of involuntary muscle contraction that results in shaking or trembling. It is a common symptom of movement disorders that affects a large number of people [1]. There are various types of tremors, but the most prevalent ones are essential tremors (ET) and Parkinson’s tremors (PT). ET affects 0.9% of the population, whereas PT is estimated to affect 0.3% of individuals under the age of 60, increasing to 1% in those over 60 years old [2]. ET is characterized by rhythmic body movements that have a frequency of 4–7 Hz [3]. On the other hand, PT typically affects non-active body parts and tends to worsen when the limbs are straightened and placed in a resting position [4].

Tremors can significantly impact an individual’s quality of life by affecting their ability to perform everyday activities such as eating, writing, and using a phone. These challenges can lead to feelings of frustration and a loss of independence. Therefore, it is crucial to invest in research and technological support to develop effective treatments, Rehabilitation, and support technologies that can alleviate the effects of tremors.

Pharmacotherapy has long been the conventional approach for managing tremors in patients with ET and PT, but its effectiveness is often hindered by various factors, including drug-drug interactions, intolerable side effects, and inadequate therapeutic response. For example, dopaminergic drugs, which are commonly prescribed for PT, only alleviate tremors in about half of the patients [5]. Similarly, anticonvulsants such as gabapentin and paracetamol, as well as beta-blockers and benzodiazepines, which are often used for ET, have limited success rates, particularly for moderate to severe cases [6,7,8]. Surgical interventions, such as deep brain stimulation, thalamotomy with radiofrequency, radiosurgery, and focused ultrasound, have shown limited long-term efficacy, which makes them a less popular option among patients [9]. Furthermore, many individuals with mild tremors view medications as detrimental to their health, which leads them to abandon treatment, while those with mild to moderate tremors are generally hesitant to undergo surgery. For patients with severe and advanced tremors, surgery is often not a viable option due to the risks and complications associated with the procedure. Therefore, non-pharmacological interventions, such as regular exercise, must be considered as an alternative treatment approach.

Rehabilitation is an important adjunct to clinical medical treatment for tremors, particularly in patients with mild to moderate disease who are reluctant to undergo surgery and can benefit from physical therapy programs. Various approaches have been studied, including flexibility and strength training. Sajjad Farashi et al. [10] conducted a meta-analysis of a large body of literature and found that exercise significantly reduced tremor in patients with Parkinson’s disease, with hand exercises showing promise for reducing distal limb tremors. Strength training involves applying resistance to the limbs of patients with tremors to stimulate neural control of muscles, and studies by G. Sequeira et al. [11], J. Kavanagh et al. [12], and M. Bilodeau et al. [13] have shown that a generalized upper limb strength program has the potential to improve stability and flexibility in patients with PT or ET. However, these studies also found that patients may experience fatigue during training and that functional capacity does not always improve after training [13]. Flexibility training is another useful method for improving execution and control in patients with tremors and gradually enhancing their self-confidence. Mona Kadkhodaie et al. used eccentric-based rehabilitation training and found a significant reduction in the amplitude of resting tremor after exercise in the intervention group, although the study had limitations in terms of assessing tremor fluctuation [14]. N.E. Vance et al. [15] used yoga to rehabilitate patients with primary tremor and demonstrated improvement in tremor assessment scales after an eight-week intervention. W. Chung et al. [16] provided behavioral relaxation training to ET and PT patients and found that regular relaxation training can reduce the effects of tremor, but noted the limitations of general rehabilitation training due to a lack of long-term follow-up. H. Rajalin [17] found that home rehabilitation training can improve not only the physical function of patients with Parkinson’s disease but also their activities of daily living, but noted the lack of interventions available for use at home by patients with PT. Overall, rehabilitation has the potential to improve the quality of life of patients with tremors, but further research is needed to develop effective and accessible interventions for all types of patients. In order to promote daily rehabilitation for Parkinson’s patients and reduce the pressure on rehabilitation institutions, Xiangya Hospital in China has developed a video exercise program called Yapa-PBGA (Yapa Parkinson balance and gait aerobics). This exercise program increases muscle control, improves gait and balance disorders, and upper limb flexibility in Parkinson’s patients [18]. It is cost-effective and convenient for patients to perform daily rehabilitation at home. However, its limitations are that this video format cannot provide direct rehabilitation training guidance and monitoring to patients, especially considering the uncontrollable situation of Parkinson’s patients. It is difficult to achieve the ideal training effect through this training method.

Virtual reality or augmented reality technology can overcome the limitations of video-based rehabilitation training methods by providing interactive visuals and personalized guidance, thus increasing patient engagement and motivation during the rehabilitation process. Hueso et al. [19] developed a virtual reality remote assistance system that allows therapists to create customized treatment plans and automatically record the patient’s movement. However, the system lacks flexibility and is not suitable for developing tremor rehabilitation training. J. Cornacchioli [20] studied the use of the Oculus Rift grip as a tool to detect Parkinson’s symptoms by measuring involuntary hand movements. The study tested the effectiveness of Parkinson’s symptoms and concluded that the accuracy of Oculus Rift was sufficient for measurement needs but did not develop the technology for tremor rehabilitation training. G.C. Burdea et al. [21] reviewed the advantages and disadvantages of virtual reality rehabilitation applications and summarized the benefits, including improved patient motivation, adaptive data access, online data access, and reduced healthcare costs. They also revealed the limitations of virtual reality technology in rehabilitation, including a lack of supportive infrastructure, expensive equipment, and inadequate communication infrastructure for rural tele-assistance. Jiang et al. [22]. presented a multi-category gesture recognition model that uses signals from both surface electromyography and inertial measurement units. The model aims to improve the accuracy and robustness of gesture recognition in various real-world applications, such as human-computer interaction and rehabilitation.

Augmented reality refers to the combination of digital information from a virtual world with the physical environment of the real world to create a more interactive and immersive user experience. Wang et al. [23,24] conducted a study on the application of augmented reality technology to assist tremor patients in typing on a keyboard like ordinary people. Wang et al. [25] investigated the use of projection-based augmented reality technology called “Extend Hand” to help tremor patients directly interact with remote-controlled home appliances. Compared to virtual videos, augmented reality can provide more intuitive and specific rehabilitation training guidance and monitoring. By adding digital elements to real-life scenes, patients can better understand, simulate, and practice rehabilitation training skills, thus enhancing their rehabilitation outcomes. In addition, augmented reality technology can provide more personalized rehabilitation training plans and feedback, which can enhance patient engagement and motivation, leading to more active participation in rehabilitation training. Therefore, augmented reality technology has great potential to become an effective auxiliary tool for rehabilitation training. Aditya Pillai et al. [26] developed an innovative mixed reality rehabilitation tool specifically designed for upper limb injuries that utilized HoloLens 2 technology. The tool overlaid digital elements onto the real-world scene through the augmented reality feature of HoloLens 2, providing personalized rehabilitation training guidance and monitoring to help patients with upper limb injuries regain their function. However, the expensive price of HoloLens 2 limits its adoption for rehabilitation training that can be conducted at home by users. The Oculus Quest 2 by Meta is a milestone virtual reality headset that allows for free movement without the constraints of cables and is available at an affordable price. Its passthrough feature can be developed into an AR device. Additionally, the built-in controller Oculus Touch, featuring infrared emitters and inertial sensors, has been proven to be accurate and robust in detecting hand movements [27,28].

Considering the lack of a specialized augmented reality system for rehabilitation training specifically designed for patients with tremors and the need to improve the effectiveness and experience of rehabilitation training that can be conducted at home by patients with tremors, including one-on-one demonstration, posture guidance, and monitoring of training progress, we developed a low-cost augmented reality tremor rehabilitation training system using Oculus Quest 2. This study used Optitrack to record standard Yapa-PBGA movements and incorporated them into the system, creating a one-on-one virtual model for rehabilitation training. By designing natural interaction logic in the system, individuals with tremors can be guided to perform accurate limb movements during rehabilitation training. The patients’ actual limb movement data can also be recorded and analyzed in real-time by the system.

## 2. Pilot System

### 2.1. System Configuration

In this study, an optical see-through AR system based on Yapa-PBGA is proposed to support individuals with tremors in rehabilitation training. The system employs Oculus Quest 2 and its self-contained controller, Oculus Touch. As illustrated in Figure 1, the virtual information to assist in tremor rehabilitation is registered in a physical space visible to the user by the development environment of the “passthrough” of Oculus. Oculus Touch was used to track the hand*’*s position (posture) during rehabilitation training.

### 2.2. System Visual Presentation

In the system environment, the user is rehabilitated as if he/she were playing Frisbee. The user can not only see his/her body and hands, but also a human-like avatar, virtual targets, and visual prompting for hand positions. This avatar is designed so that it simulates the gymnastic posture of the instructor in Yapa-PBGA and is registered in the physical environment to demonstrate the physical movements of rehabilitation training (See Figure 2a). A virtual target in the shape of a Frisbee follows the rhythm and sequence of rehabilitation gymnastics and will appear at the designated location for the user to pick up to guide their movement. (See Figure 2b). A pair of virtual bubbles synchronized with the position of the Oculus Touch informs the user in real time where the hand is located. After training, the system calculates and evaluates the user’s training by comparing Oculus Touch’s real-time tracking data with standard data (See Figure 2c).

## 3. Methods

### 3.1. Yapa-PBGA Rehabilitation Posture Model

Yapa Parkinson balance and gait aerobics abbreviated as Yapa-PBGA is a video-based dexterity training gymnastics for tremor patients that is proposed by the National Clinical Research Center for Geriatric Disorder, XiangYa Hospital [16], to improve gait and balance disorders as well as upper limb dexterity. This study summarized and extracted the Yapa-PBGA rehabilitation posture model, including “side bend up (G1),” “side clap (G2),” “drop up (G3),” “sun hug (G4),” “side stretch (G5)”, “side flight (G6),” “spiral down finger (G7)”, “greeting (G8)”, and “tai chi (G9)”, for use in developing the system and conducting experimental testing (Figure 3).

### 3.2. Creation of Demonstrator Avatar

To create an immersive rehabilitation training experience that simulates the presence of a virtual demonstrator performing gymnastics in front of the user, we have designed a gymnastics demonstrator Avatar. According to the Yapa-PBGA posture model, a motion actor who underwent extensive Yapa-PBGA training was arranged to carefully choreograph and record the movements of Aatar, ensuring the accuracy of the postures.

In Figure 4, 10 motion capture devices, Opti-Track, were positioned around the Yapa-PBGA demonstrator in order to capture the demonstrator’s poses and motion positions accurately and fully. After that, the demonstrator’s postures were bound to the avatar by using Opti-Track motive software and the demonstrator’s positions were transformed to the AR system by applying the spatial coordinate transformation. In addition, the demonstrator’s sitting height, arm length, and distance from their heads were manually measured to adjust the system for different body types.

### 3.3. Guidance and Detection

To guide users to maintain standard limb postures during rehabilitation training and make the training more engaging, a natural interaction logic of catching frisbees is designed for the rehabilitation training system. Specifically, the interaction logic includes (1) The rehabilitation training system first presents the standard limb postures to guide users to maintain the correct posture. (2) Frisbees fly from a distance and reach only the hand positions corresponding to the standard limb movements. (3) Users need to catch the virtual frisbees within a specified time. (4) Users receive visual feedback from the frisbees when they successfully catch them, enhancing their perceptual experience. (5) When the user successfully catches a virtual frisbee, the system emits Vibration Feedback to enhance the user*’*s sense of achievement. (6) The actual landing position of the frisbee is dynamically adjusted according to the user’s 3D spatial coordinate position to ensure accurate landing on the user*’*s hand position. Using this natural interaction logic can promote users to maintain standard limb postures during training, improving the quality of rehabilitation training.

The user’s motion data are tracked by Oculus Touch. In light of the fact that a tremor patient’s vibration frequency is generally less than 10 Hz, according to the Shannon theorem, the actual detection frequency that was set at 20 Hz was sufficient to meet the motion data collected without any distortions. User quality of movement in each gymnastic posture is measured by comparing user motion data with Frisbee positions. As shown in Equation (1), when the Euler distance between the Frisbee and the handle is less than 100 mm, i.e., when the user’s limb is close to the specified position. The system offers the user both visual feedback on the frisbee’s disappearance and haptic feedback through vibrations.
(1)$${State}^{t}=\left\{\begin{array}{c} Standard,\sqrt{{\left(P_{x}^{t}-p_{x}^{t}\right)}^{2}+{\left(P_{y}^{t}-p_{y}^{t}\right)}^{2}+{\left(P_{z}^{t}-p_{z}^{t}\right)}^{2}}\leq 100\;m\\ Non-standard,\;Otherwise \end{array}\right.$$
where ${State}^{t}$ is recorded whether the user’s posture was standard at time t, P is the position of the virtual Frisbee relative to the helmet, and *p* is the position of the user’s hand/Oculus Touch relative to the helmet.

## 4. Experiment

### 4.1. Experimental Setting

As part of the study, experiments were conducted to test the efficacy of the proposed system in helping people with tremor during rehabilitation. To simulate patients with Parkinson*’*s disease (PT) and primary tremor (ET), participants were asked to wear a tremor simulator. Our study used a tremor simulator, shown in Figure 5, consisting of an Arduino Uno, a dual-channel muscle electrical stimulation module, and electrode pads. The Arduino Uno and the muscle electrical stimulation module used IIC communication to send a boosted electrical stimulation pulse current to the electrode patches. The electrode patches were placed on the lateral side of the participant*’*s left and right hands, 3 cm from the elbow joint and 2 cm from the wrist joint. To replicate the tremor experienced by patients with PT and ET, we used a tremor simulator to randomly apply two types of electrical pulses to the participants*’* upper limbs. One signal caused the limb to tremble at a fixed frequency of approximately 5 Hz, while the other allowed the limb to oscillate voluntarily at a frequency of 4 to 7 Hz. We assessed their limb tremor status with Oculus Touch before each experiment to ensure that participants*’* involuntary limb tremors met tremor criteria.

The experiment will record participants’ motions in the video and AR environments, respectively. To objectively compare the effects of rehabilitation training in the video and AR environments, this study further processed the sampled data and analyzed the magnitude of the body movements of the participants as they performed the rehabilitation postures in Figure 1 in both environments as shown in Equation (2).
(2)$$M=\sum_{i=1}^{n}\sqrt{{\left(\sqrt{x_{i}^{2}+y_{i}^{2}+z_{i}^{2}}-\sqrt{x_{i-1}^{2}+y_{i-1}^{2}+z_{i-1}^{2}}\right)}^{2}}$$
where M is the magnitudes of body movements, n is the number of sampling, x,y,z are the position of the two Oculus Touch (hands) relative to the helmet.

An in-depth survey of the participants’ rehab experiences was conducted through the use of the following questionnaire. The psychological feelings in relation to questions from Q1 to Q12 were rated based on a seven-point Likert scale, ranging from 0 (strongly disagree) to 7 (strongly agree).
Q1: I feel comfortable doing rehabilitation training in this environment.Q2: I feel interested in rehabilitating in this environment.Q3: I find it easy to do rehabilitation in this environment.Q4: I think I can tolerate rehabilitation in this environment.Q5: I feel unburdened by rehabilitation in that environment.Q6: I feel that the body movements are standard in this environment.Q7: I feel like I can follow the pace of rehabilitation training in this environment.Q8: I feel like the environment can guide my body movements.Q9: I feel as if rehabilitation in that environment would have good results.Q10: I am satisfied with my rehabilitation training in this environment.Q11 (SO): Sense of Ownership: I felt as if I was touching the virtual object directly with my hands, forgetting the existence of the handle.Q12 (SA) SA: The human-computer interaction is easy to control without a sense of dissonance in the AR rehabilitation system.

### 4.2. Experimental Procedure

Firstly, we invited a physically healthy Yapa-PBGA demonstrator to perform a set of standard exercises. Throughout the entire training routine, the demonstrator wore an Oculus Touch, and the 3D positional coordinates of his upper limb movements were fully recorded. Subsequently, we extracted data from the movements in the postures shown in Figure 1 as the control group for the experiment. Next, we invited 12 healthy individuals aged between 20 and 30 to participate in the intervention study. The participants completed rehabilitation training under both the Augmented Reality (AR) condition and the video condition, in a randomized order. The tremor simulator worn by the participant was calibrated before the experiment to ensure that the participant’s body trembling reached the frequency of ET or PT. Participants were required to perform a 5-min adaptation exercise to familiarize themselves with the task. In the experimental phase, each participant completed the rehabilitation task including nine rehab postures (Figure 1) using a tremor simulator and Oculus Touch. The participant answers the questionnaire after a sectional experiment and is given a rest to prepare for the next experimental condition.

## 5. Results

As a way of clearly describing, analyzing, and comparing the two rehabilitation modalities in the following sections, “Video” and “AR” represent the video and AR rehabilitation conditions, respectively, “Q1” to “Q10”, the experience investigations, and “G1” to “G9” (“L” is the left body, “R” is the right body), the rehab postures.

A compared T-test was conducted to compare the magnitude of the mean body movement of participants making G1 to G9 under the “video” and “AR” conditions (see Figure 6). There was a significant effect on the mean body movement magnitude at the ***p*** < 0.05 level for G2, G3, G4, G5, G7, and G9. In groups G1 and G8, significant differences were commonly found in the L group, but not found in the R group. The results revealed that for most rehab postures, the magnitude of human motion in the AR condition was significantly greater than that in the video environment. As shown in Figure 6, the red lines indicate the actual magnitude of movement of the control group in making each rehab posture, respectively. From the results of the experiment, the AR group was remarkably closer to the control group compared to the video group. To sum up, the proposed AR method can help individuals with tremors make Yapa-PBGA postures that achieve a magnitude of body movements close to that of a standard demonstrator and can effectively promote the quality of movements in rehabilitation training for them in comparison to the video.

Some psychological aspects of the survey further evaluated participants’ experience with “Video” and “AR” rehabilitation training. A compared T-test was used to analyze the difference in emotional experience between the conditions of “Video” and “AR” through the questions from Q1 to Q5. The results are given in Figure 7. Both conditions showed statistically significant differences with *p* < 0.01. The results showed that the participants felt comfortable and easy doing rehabilitation, did not perceive the physical burden, and were able to tolerate the intensity of the rehabilitation training. In the analysis, we compared the differences between the two conditions on the basis of Q6 to Q8 and found significant differences at *p* < 0.01. The result shows that the participants were more likely to be guided by rehabilitation training. Thus, the participants were more confident in performing standard rehabilitation training movements in the AR environment. There are no significant differences between the conditions in Q7, however, as the participants generally rated it very well (See Figure 7), it means participants could follow the pace of rehabilitation training under the AR condition. The analysis of Q9 and Q10 utilized a comparative T-test, revealing that the participants generally expressed satisfaction with the AR environment and preferred it as a rehabilitation training setting.

In addition, the proposed methods were evaluated from the perspective of a sense of ownership and agency. The sense of ownership (SO) and the sense of agency (SA) are two central aspects of bodily self-awareness; the sense of ownership would be considered a direct perceptual experience, while the sense of agency is the sense of controlling and causing the body to act through volition, two important properties of operational logic in human-computer interaction. As we calculated the mean and standard deviation on the questionnaire of Q11 (SO) and Q12 (SA), we found a mean of 5.5 with a variation of 0.48, and a mean of 5.7, with a deviation of 0.3. This implies that participants are prone to the illusion that they are touching virtual objects, that the interaction can be easily controlled, and that users do not experience a strong sense of dissonance in the AR environment.

## 6. Discussion

Rehabilitation is an important way to reduce the severity and progression of tremors, and Yapa-PBGA provides a low-cost video rehabilitation program that can be used by patients at home. However, this program has limitations in providing face-to-face guidance for patients and does not track the quality of movements during rehabilitation. To address these limitations and better assist tremor patients with their rehabilitation at home, a low-cost AR system has been demonstrated to create an immersive rehab experience. This system simulates an instructor demonstrating gymnastics in front of the user, while also guiding the user’s body movements during the rehabilitation process.

To evaluate the effectiveness of the proposed AR system in supporting individuals with tremors to achieve better results in Yapa-PBGA gymnastics, experiments were conducted to compare and analyze the magnitude of body movements of individuals with tremors in the video and AR environments, as well as that of the control group. The magnitude of movement is an objective measure of how well the movement achieves the correct posture, and the greater the magnitude, the better the rehabilitation will be. In most rehabilitation postures, participants demonstrated significantly greater magnitudes of limb movement in the AR environment, approaching the demonstrator’s level, compared to the video environment. However, no statistically significant differences were observed in G1 (R) and G8 (R). Further analysis revealed that this was due to the fact that these two groups used poses involving hands on the waist, which are easy to assume and least influenced by experimental conditions. Therefore, G1 (R) and G8 (R) were not considered in the final experimental conclusion. It can be concluded that the proposed AR method is effective in improving the quality of movement for individuals with tremors. User experience surveys showed that participants felt more relaxed, comfortable, and engaged when using the proposed AR system compared to the video environment. The proposed AR system was also effective in providing rehabilitation guidance to individuals with tremors. Although there was no significant difference between the AR environment and the video environment in Q7, the actual evaluation revealed that participants could follow the pace of rehabilitation training under AR conditions. Additionally, an analysis of the sense of ownership and agency of participants with tremors found that the AR system was effective in providing them with a natural rehabilitation experience without causing dissatisfaction.

## 7. Conclusions

This study aimed to develop a low-cost augmented reality rehabilitation training system that would enable tremor patients to receive training at home with guidance. We created a set of rehabilitation posture models by extracting basic movements from Yapa-PBGA and established a one-to-one avatar demonstration training action through three-dimensional reconstruction and virtual mapping. To make rehabilitation training more engaging, we designed a natural interaction logic using a frisbee interaction mode to guide rehabilitation trainees’ posture and provide necessary visual and tactile interaction feedback. We evaluated the effectiveness of our proposed tremor rehabilitation system by conducting simulated experiments and comparing it with traditional video rehabilitation methods. The results showed that our system significantly improved the movement quality of tremor patients while providing a more relaxed, comfortable, guided, and controllable rehabilitation experience than video rehabilitation methods. However, due to the limited number of patients tested, further optimization and discussion are still necessary for clinical settings. Long-term tracking and evaluation of the system’s effectiveness are also essential. Therefore, we plan to continue improving the system and expanding the sample size in future research to further validate its effectiveness and feasibility.

## Figures and Tables

**Figure 1 sensors-23-03924-f001:**
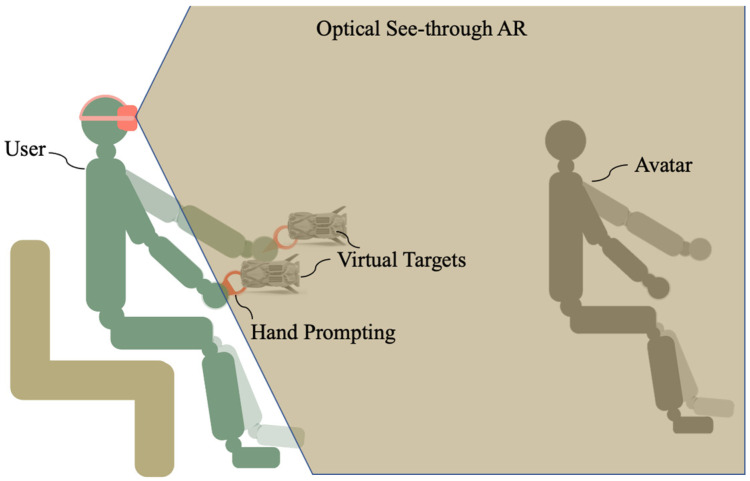
Tremor rehabilitation system using optical see-through AR.

**Figure 2 sensors-23-03924-f002:**
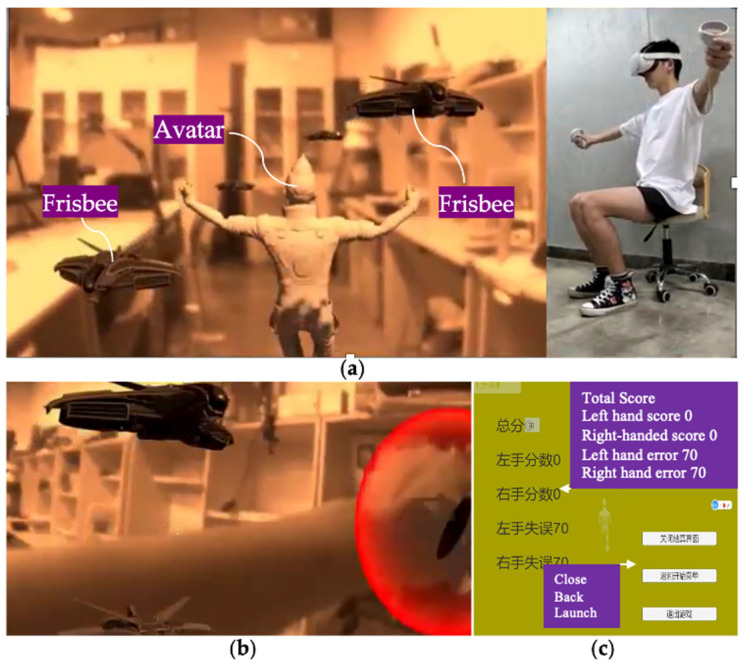
Under the proposed system, (**a**) An avatar, virtual targets, and physical environment can be observed, and (**b**) a user’s hand can be seen and promoted. (**c**) an interface appears at the end of the training, presenting the system’s evaluation of a user’s rehabilitation.

**Figure 3 sensors-23-03924-f003:**
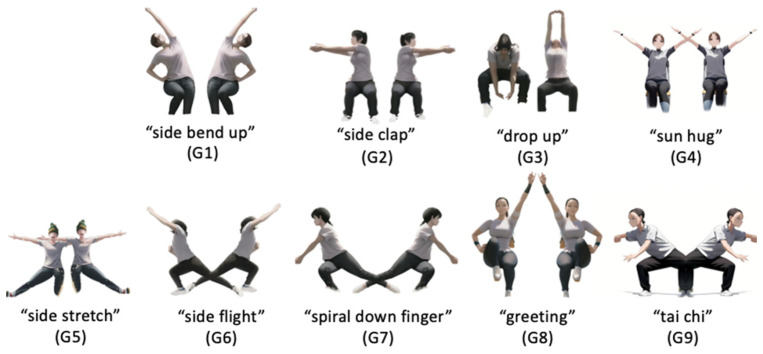
Yapa-PBGA Rehabilitation Posture Model.

**Figure 4 sensors-23-03924-f004:**
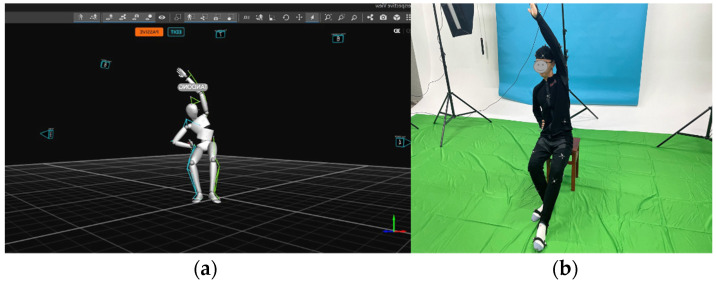
(**a**) The demonstrator wore a marked suit and demonstrated standard Yapa-PBGA gymnastics, and (**b**) his posture and movement were recorded with Opti-Track Motive.

**Figure 5 sensors-23-03924-f005:**
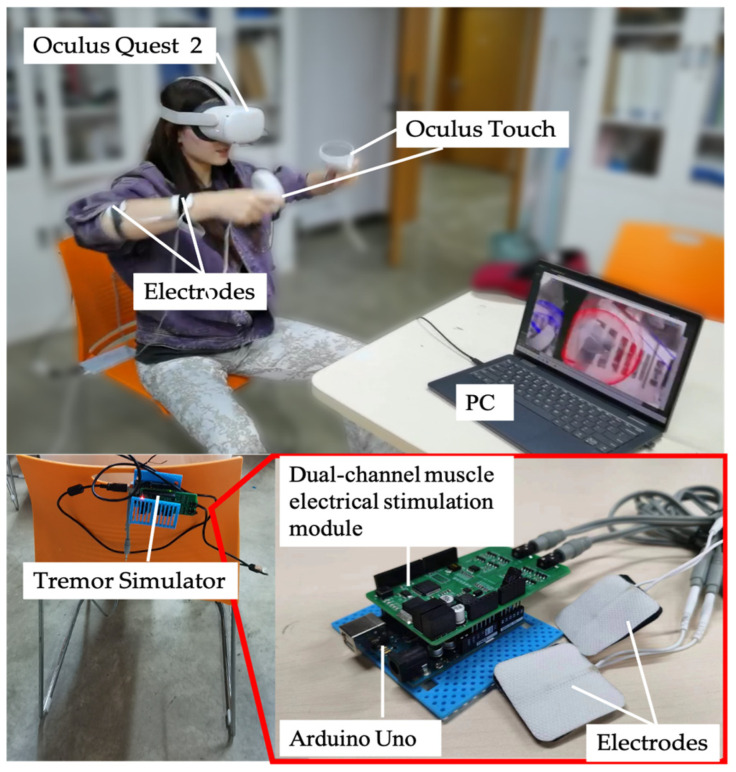
A participant is performing a rehabilitation task with a simulated trembling by a simulator.

**Figure 6 sensors-23-03924-f006:**
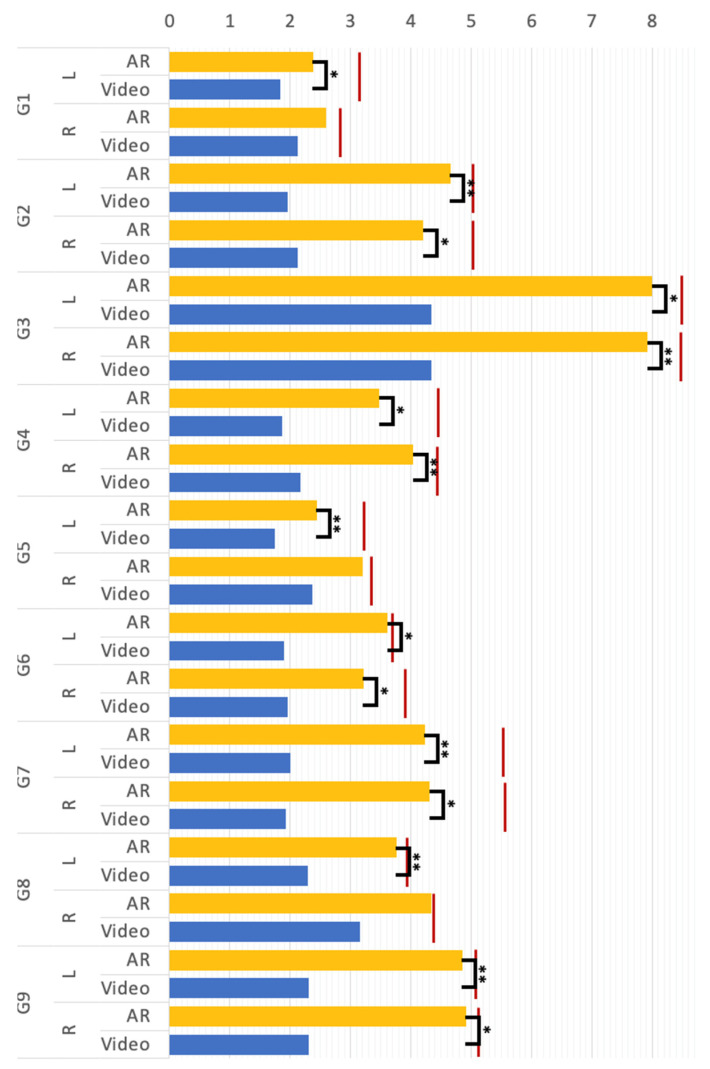
The mean body movement magnitude (** *p* < 0.01, * *p* < 0.5); G1 to G1 represent the rehab postures, respectively; “L” and “R” denote the left and right body part; *“*Video*”* and *“*AR*”* represent the video and AR rehabilitation conditions; the red lines are the value of the control group.

**Figure 7 sensors-23-03924-f007:**
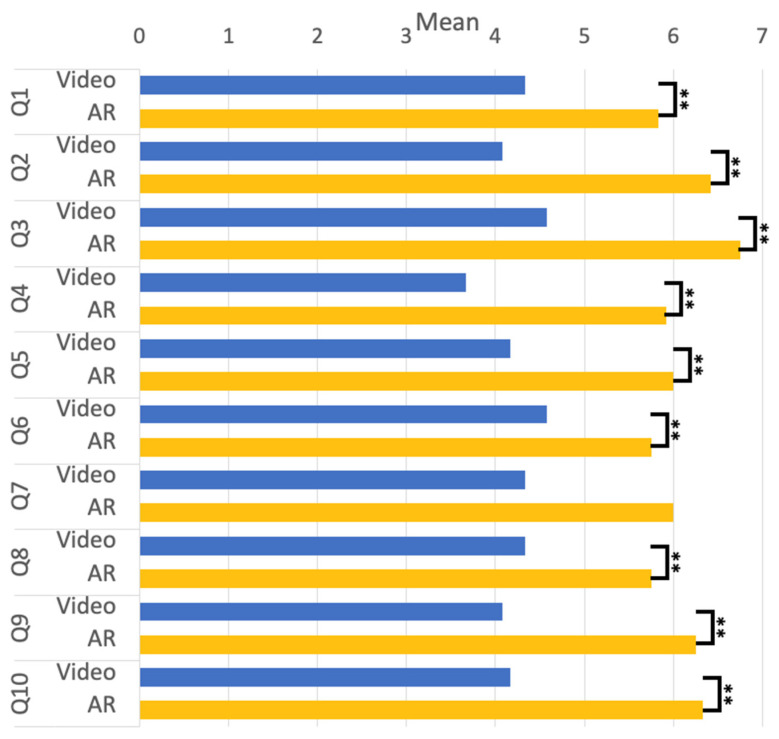
The mean score of the Q1 to Q10. (** *p* < 0.01, * *p* < 0.5).

## Data Availability

Not applicable.

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
