# Peer review of "Supporting Tremor Rehabilitation Using Optical See-Through Augmented Reality Technology"

_sensors, 2023, doi:10.3390/s23083924_

Round 1

Reviewer 1 Report (New Reviewer)

  In this work author propose method for AR rehabilitation training which is available to patients at home and which is based on the "passthrough" technology of Oculus and Yapa-PBGA.     Quality of all Figures must be significantly improved.     I wonder why experiments where only simulated on patients with PD and ET and why experiments where not conducted on real patients?   

Author Response

Dear Reviewer,

Thanks for taking the time to provide us with such valuable comments on our paper.

 We hope that we have answered all your questions. 

Reviewer 2 Report (New Reviewer)

1-      The title is inappropriate, I suggest you reconsider it.

2-      The abstract is inadequate and fails to convey the significance of the research. The beginning sentences are poor and do not accurately describe the objective of this investigation. Furthermore, it features long, dull sentences. I propose scientifically rewriting the abstract, with the following elements included. The overarching goal of the article and the research issues you looked at should be brief. The study's fundamental layout. Significant discoveries or trends made as a result of the research. finally, a succinct breakdown of your analyses and findings.

3-      The introduction is uninteresting, disjointed, and confusing. It's essential to concentrate on the introduction, convey the events in order, and be readable. Long sentences should not be used since the intended meaning is lost. Consequently, I recommend rewriting the introduction professionally while taking the following to answer the following questions:

Q1: How can you evaluate the presented results according to other studies?’ Prove that the literature review lacks such a study by more modern references.

Q2: ‘What is the importance of the presented paper?’

Q3: What is the main challenge and issues in this study?

Q4: What is the criticism and gap analysis for academic literature that attempts to provide a solution?’

Q5: What are the recommended solutions for such challenges and their issues?’

Q6: What are the present study's implications, contributions, and novelty?’

4-      The "Overview of the Proposed Methodology" section is uninteresting, disjointed, and confusing. It's essential to concentrate on the introduction, convey the events in order, and be readable. Rewrite this section and try to list (numbering) the steps to be clearer. Use a flow chart to explain the proposed approach.

5-      Any comparative analysis to testify that this study is more advanced than others? Discuss similar paper

6-      What are the defects of the method, and what are the methods that must be followed to reduce these defects?

7-      The discussion of the results is not satisfactory. Therefore, it is required to discuss the results scientifically to clarify claims and outcomes.

8-      rewrite the conclusion and consider the following comments:

-   Highlight your analysis and reflect only the important points for the whole paper.

-   Mention the benefits.

-   Mention the implication at the last of this section.

9-      The paper contains errors and typos. Remove the replicated sentences from the whole article and correct typos.

10-   The English level is low. I suggest proofreading by a specialist agent.

This paper is interesting and valuable, but major revisions may be necessary. Please carefully revise my comments.

Author Response

Dear Reviewer,

Thanks for taking the time to provide us with such valuable comments on our paper.

 we hope that we have answered all your questions. 

Reviewer 3 Report (New Reviewer)

It is no doubt that AR is a good solution for rehabilitation training. However, this paper does not provide technical innovation, e.g., AR algorithm improvement, rehabilitation system improvement. Therefore, I hardly to evaluate the scientific contributions.

The paper is more like an applied science article instead of science paper.

Author Response

Dear Reviewer,

Thanks for taking the time to provide us with such valuable comments on our paper.

 We hope that we have answered all your questions. 

Reviewer 4 Report (New Reviewer)

The subject of your paper is of great interest in developing original AR based systems for tremor rehabilitation. Your paper represents a useful work in this field. Please consider following suggestions:

- page 1, line 38, you introduce the notations ET for essential tremor and PT for Parkinsons’s tremor, but during the next lines you use abbreviation PD (which is used for Parkinson’s disease) ; please correct this first paragraph;

- page 3, 134: please write the postures: side step/side stretch (G5) and side flight (G6) according to the Figure 1;

- page 3, line 137: please correct the Figure 1 caption (is the same like Figure 2 caption);

- please improve the quality of all figures;

- although the Introduction presents a review concerning the essential tremor and Parkinson’s tremor, medical interventions (pharmacotherapy, surgery) used to manage tremors in patients, rehabilitation training and VR and AR potential in rehabilitation procedures as well, a comparison with similar systems might be of interest for researchers; please compare the proposed version and presented results, with similar AR tremor rehabilitation systems and  their performances; thus, the review and assimilation of existing literature can be improved;

- the main problem that I have identified refers to the fact that the proposed system was tested by healthy users instrumented with tremor simulators, not by real users; please give more details concerning your future research and the extension of the studies, based on the results presented in this paper, but with a larger number of real users;

- references [1] and [5] seem to be the same, please correct;

- minor English corrections are required.

Author Response

Dear Reviewer,

Thanks for taking the time to provide us with such valuable comments on our paper.

 We hope that we have answered all your questions. 

Round 2

Reviewer 2 Report (New Reviewer)

Dear Authors

This manuscript is interesting and contains important information for readers. I can recommend this paper for publication with no modifications needed

Regards

This manuscript is a resubmission of an earlier submission. The following is a list of the peer review reports and author responses from that submission.

Round 1

Reviewer 1 Report

Dear Authors,

After reading it, I make a few notes:

Interesting subject, involving high technology

Summary: I missed the study methodology;

Introduction: good contextualization, well founded;

Methodology: clear, objective and detailed enough for understanding the study; I just missed the justification of the choices, I think they are more present in the Introduction than in the Methodology.

Experiment: it was presented in a clear and objective way;

Result: I missed the discussions of the results, as is normal in most areas; there is only the presentation of experiments and results;

Conclusion: brief, also did not present discussions.

Author Response

Dear reviewer

We would like to thank the reviewer for taking the time to provide us with such valuable comments about our paper. Based on your comments, we have revised the paper and added content. Please refer to the document of "Response Letter 1.pdf". We hope that all questions have been answered and would appreciate any suggestions you may have.

Reviewer 2 Report

This paper deals with an exciting topic. The article has been read carefully, and some minor issues have been highlighted in order to be considered by the author(s).

#1 What is the motivation of this paper?

#2 What is the contribution and novelty of this paper?

#3 What is the advantage of this paper?

#4 Which evaluation metrics did you used for comparison?

#5 It would be good if security domains for the deep neural network would be reflected in the related work such as “Defending Deep Neural Networks against Backdoor Attack by Using De-trigger Autoencoder” 

Author Response

Dear reviewer

We would like to thank the reviewer for taking the time to provide us with such valuable comments about our paper. Based on your comments, we have revised the paper and added content. Please see the attachment. We hope that all questions have been answered and would appreciate any suggestions you may have.

Reviewer 3 Report

This is a very interesting study presenting a rehabilitation system using optical see- 16 through augmented reality. The proposed system seems to have the potential to contribute to both academia and the treatment in practice. The overall readability is strong. Here are just some minor suggestions for the authors.

1. A more comprehensive background is much preferred to highlight the novelty of the proposed system and its potential implications.

2. Figure 2 (c) is presented in Chinese. It would be great if the authors can change it to an English version or explain the information in English.

3. It would be ideal if the authors can provide some more detailed information in Figure 3. 

Author Response

(The authors gave the same response as above.)

Reviewer 4 Report

With the limited amount of research literature available for the focus of the manuscript, once the revisions are completed, this has the potential to add relevant material to the profession. A thorough description of the design of this study is warranted, allowing the reading audience to determine if the study is transparent and reproducible. Once that has been included, readers can determine the significance of the study's contribution. Conclusions could be stronger based on proposal and results. Mentioned comparing tests and t-test, but did not see the table in the article or as a supplement.

Author Response

(The authors gave the same response as above.)

Round 2

Reviewer 2 Report

The authors have answered some of my previous questions/suggestions. However, the corresponding improvements are unclear and very difficult to follow in the revised manuscript. In a nutshell, the argued innovations in terms of theory (design and analysis) are minor and/or questionable.

Author Response

Dear reviewer

We would like to thank the reviewer for taking the time to provide us with such valuable comments about our paper. Based on your comments, we have revised the paper and added content. Please refer to the attachment. We hope that all questions have been answered and would appreciate any suggestions you may have.

Reviewer 4 Report

The authors list this as an experimental study. This implies a control group, in addition to the interventional application group. I did not see a second group. Nor do I see this study on 'normal' 20-30 year old individuals helpful for those with Parkinsonian tremors or other neurological deficits that exhibit tremor-like body movements. The conclusions are still weak.

Author Response

(The authors gave the same response as above.)

Round 3

Reviewer 4 Report

Line 20-23: Please review experimental reports, their protocol, and method of dissemination of results and conclusions. “Experimental reports follow a general to specific, to general pattern. Your report will start out broadly, in your introduction and discussion of the literature; the report narrows, as it leads up to your specific hypotheses, methods, and results. Your discussion transitions from talking about your specific results to more general ramifications, future work, and trends relating to your research.” (https://owl.purdue.edu/owl/subject_specific_writing/writing_in_the_social_sciences/writing_in_psychology_experimental_report_writing/experimental_reports )  

 Line 17-18: Read the intent of a research proposal. Basic requirements of a proposal might be helpful to read. Discerning between a proposal (which this reads), and an experiential report, is necessary for your manuscript. (Sudheesh K, Duggappa DR, Nethra SS. (2016). How to write a research proposal?. Indian J Anaesth (60)9, 631-634. https://doi.org/10.4103/0019-5049.190617 )

Aims of Sensors

Sensors (ISSN 1424-8220) provides an advanced forum for the science and technology of sensors and their applications. It publishes comprehensive reviews and regular research papers. Our aim is to encourage scientists to publish their experimental and theoretical results in as much detail as possible. The full experimental details must be provided to reproduce the results.

I am still unable to determine the exact intent of your manuscript, as an experimental study or a research proposal. These represent two different means of conducting the study or the need for research through a proposal. This will need further work for clarity and transparency for the reading audience.

 Line 27-39: Developing or describing the AI intervention with participants in the 20-30 age range with no underlying neurological tremors is lacking testing on the intended individuals (described as over 60 or 65 with Parkinson’s disease).

Line 44-49: duplicate. Delete

Line 53: What medical limitation? Describe it.

Line 115-121: This section fits better in conclusions

Line 147-157, 283-285: How easy will this process of AI be for elderly folks more than age 65 to use, especially with severely compromised tremulous limbs?

Line 159-161: rephrase, hard to read and follow

Line 196-243: Experiment: describe the inclusion criteria for each group of participants, if this is a true experimental model. Be very clear who is the control group and interventional group.

Line 261: where is this test?????? There is no visual table in this article. Figure 6 does not represent the standard t-test.

Line 293, 300-301: what are you saying? Please clarify.

Line 314-316: show the experimental comparisons, analysis, results between the groups

Line 320-321: what individual groups??? Describe

Line 321-324: where do you want this information, results or conclusions?

Line 325-332: duplicate, delete

EQUATOR: Enhancing the QUAlity and Transparency Of health Research. Reporting guidelines for main study types.  https://www.equator-network.org/